# Generative Location Modeling for Spatially Aware Object Insertion

## Abstract

Generative models have become a powerful tool for image editing tasks, including object insertion. However, these methods often lack spatial awareness, generating objects with unrealistic locations and scales, or unintentionally altering the scene background. A key challenge lies in maintaining visual coherence, which requires both a geometrically suitable object location and a high-quality image edit. In this paper, we focus on the former, creating a *location model* dedicated to identifying realistic object locations. Specifically, we train an autoregressive model that generates bounding box coordinates, conditioned on the background image and the desired object class. This formulation allows to effectively handle sparse placement annotations and to incorporate implausible locations into a preference dataset by performing direct preference optimization. Our extensive experiments demonstrate that our generative location model, when paired with an inpainting method, substantially outperforms state-of-the-art instruction-tuned models and location modeling baselines in object insertion tasks, delivering accurate and visually coherent results.

## 1 Introduction

Explicit modeling of object locations has recently proven to be an effective strategy for generating complex scenes. Methods that use this strategy often separate the tasks of determining *where* objects should be placed and *what* those objects should look like (Feng et al., 2024a; Phung et al., 2024; Cho et al., 2024; Lian et al., 2023). This two-step approach typically involves generating a spatial layout and then conditioning an image generation model on this layout to create the final image. The success of these approaches has brought renewed attention to location modeling as a key component for achieving realistic and coherent scene generation.

We explore explicit location modeling for *object insertion*, which involves determining where new objects can be placed in a given scene. This is an important task with applications in generative data augmentation (Zhao et al., 2023; Kupyn & Rupprecht, 2024; Fang et al., 2024), virtual reality (Park et al., 2005) and robotics (Cheong et al., 2020). Given a scene image and an instruction specifying an object to be added, object insertion aims to place a new object into the scene with a realistic appearance and geometry, while preserving the background and other objects intact. Unlike generating a full scene layout (Gupta et al., 2021), object insertion must account for the rich contextual information provided by the image, which also imposes strong constraints on where objects can and cannot be placed.

Current state-of-the-art object insertion methods rely on instruction-tuned image editing (Brooks et al., 2023; Zhao et al., 2024; Zhang et al., 2024b; Wasserman et al., 2024), which requires models to jointly learn the realism in both appearance (what) and spatial placement (where). This is a difficult task that requires large-scale object insertion datasets that are inherently hard to construct. Despite training on such datasets, existing approaches often favor realism in appearance over placement, generating objects in unrealistic locations, replace or destroy existing objects, and create unintended changes in unrelated areas, as seen in Figure 1 (a).

In this work, we propose a two-stage object insertion pipeline that first identifies the object location using a dedicated *location model*, then generates the object locally in the appointed location using an inpainting model, as illustrated in Figure 1 (b). Object placement datasets (Liu et al., 2021; Wasserman et al., 2024) suitable for training such a model have recently been released. However, as it is

Figure 1: Our proposed pipeline for object insertion (b), in contrast to instruction-tuned methods (a). We use a pretrained inpainting model by providing it with plausible locations for insertion.

unfeasible to manually label all possible bounding boxes where objects can be placed, these datasets are inherently sparse, providing annotations for less than 1% of possible locations. Additionally, there may be multiple realistic locations for any given image, making location modeling a one-to-many problem. State-of-the-art object placement models handle these issues by either assuming unlabeled areas as implausible locations (Lin et al., 2018; Tripathi et al., 2019; Zhang et al., 2020) or crafting custom loss functions (Niu et al., 2022; Zhu et al., 2023). However, these strategies risk penalizing unlabeled positive locations and are also highly sensitive to annotation sparsity, limiting their scalability across datasets.

To overcome these challenges, we approach the problem from a generative perspective, as training a generative model only requires access to *samples* from the target distribution. Specifically, we represent the input image and the object class as a sequence of tokens and use an autoregressive transformer model (Vaswani et al., 2017; Radford et al., 2019) to iteratively decode bounding box coordinates of plausible object locations. Furthermore, we can also fully leverage any additional negatively labeled locations by treating pairs of positive and negative labels as a preference dataset, where positive locations are preferred over negative locations. This perspective allows for direct preference optimization (Rafailov et al., 2024) on the location model, which further enhances the accuracy of the locations.

We empirically observe that given a precise location of the object to be generated, off-the-shelf inpainting models outperform state-of-the-art instruction-tuned object insertion models. Moreover, we observe that any inaccuracy in estimating the object location can significantly degrade the image quality, highlighting the role of location modeling as the critical component of the object insertion pipeline. Our location model produces high-quality locations, allowing it to outperform both instruction-finetuned models in object insertion and existing location models in terms of positional accuracy. We further validate the effectiveness of our approach in a user study.

Summarizing our main contributions are as follows:

- We propose a two-stage object insertion approach, that overcomes the limitations of instruction-tuned editing with an explicit location model.

- We effectively handle sparsity in location annotations with a generative approach: an image-conditioned autoregressive transformer that models bounding box locations.

- We can leverage available negative annotations following our generative formulation using direct preference optimization, further improving the accuracy of our location model.

- Through extensive experiments and a user study, we demonstrate that our approach achieves state-of-the-art performance in location modeling, and significantly outperforms instruction-tuned image editing models in object insertion tasks.

## 2 RELATED WORK

**Instruction-based Image Editing.** InstructPix2Pix (Brooks et al., 2023) introduced an approach that finetunes Stable Diffusion (Rombach et al., 2021) to interpret text instructions, trained with a dataset of paired images before and after specific edits. Subsequently, other methods (Zhang et al., 2024b; Hui et al., 2024; Zhang et al., 2024a) have released similar datasets for instruction-tuning, capable of changing the style and content of a given image.

Only recently has there been an emphasis on adding objects using text instructions, supported by datasets that provide paired images where specific objects have been artificially removed by inpainting (Wasserman et al., 2024; Zhao et al., 2024). One downside of these datasets is that they contain inpainting artifacts, which can cause models trained on them to replace existing objects or alter backgrounds. Additionally, since these models are typically trained to regenerate the entire image, they often introduce unintended changes to the scene. In contrast, our approach decouples the object insertion process by using a dedicated location model for determining placement and an inpainting model for rendering. This factorization allows for more control and precision in object insertion.

**Object Placement.** Traditionally, object placement has relied on copy-pasting an object segment by simply determining its location and scale (Zhang et al., 2020; Zhu et al., 2023; Zhou et al., 2022; Tripathi et al., 2019; Niu et al., 2022). However, this approach is not ideal for inserting objects into background images, as it requires the user to prepare an image of the object that fits seamlessly into the scene purely by placement.

To avoid relying on object segments, some approaches predict locations from class labels by querying a classifier (Dvornik et al., 2018) on random bounding boxes, categorizing them as either plausible or implausible locations for the given class. To enable training a discriminative model, unlabeled locations are typically treated as negatives or implausible locations. Such an assumption is not always valid in object placement, as the lack of annotations for specific locations does not necessarily indicate they are implausible. Therefore, penalizing them may result in an inaccurate location model. To overcome this limitation, we instead propose a generative approach that requires only samples from the target distribution, and does not make any assumption about unlabeled locations and only requires samples of the target distribution (*i.e.*, positive locations).

**Layout Generation.** Another related category of work is focused on scene layout generation, usually via a generative model of bounding box locations (Jyothi et al., 2019; Gupta et al., 2021; Chai et al., 2023; Inoue et al., 2023) or segmentation maps (Lee et al., 2018). Such layouts are often used as a condition for image generation models such as GLIGEN (Li et al., 2023) to generate complex scenes (Feng et al., 2024a; Phung et al., 2024; Cho et al., 2024; Lian et al., 2023; Gani et al., 2024; Feng et al., 2024b). Unlike these full-layout generation approaches, our goal is to insert an object into an existing scene. Having access to the background image fundamentally changes the nature of the task. On one hand, the background image provides valuable context for realistic object placement, but on the other hand, it imposes strong constraints on where objects can be placed. We therefore design a location model that integrates these contextual cues and avoids placement in unrealistic locations.

## 3 METHOD

### 3.1 GENERATIVE LOCATION MODELING

Given the distribution of image $X$, plausible object locations $Y$, and classes $C$, we frame the location modeling problem using a *generative model*, which estimates the conditional probability of the *locations* as

$$P(Y \mid X, C) = \prod_{Y_i \in Y} P(Y_i \mid X, C), \tag{1}$$

where $Y_i$ are different locations for an object of class $C$ to be placed within the image $X$. Note that unlike discriminative models $P(C \mid X, Y)$, which require labels for both positive and negative locations to classify a given location, a generative model only requires samples of positive locations.

To model this generative process, we train an autoregressive model (Vaswani et al., 2017) that sequentially predicts the bounding box coordinates of plausible locations. Specifically, each location

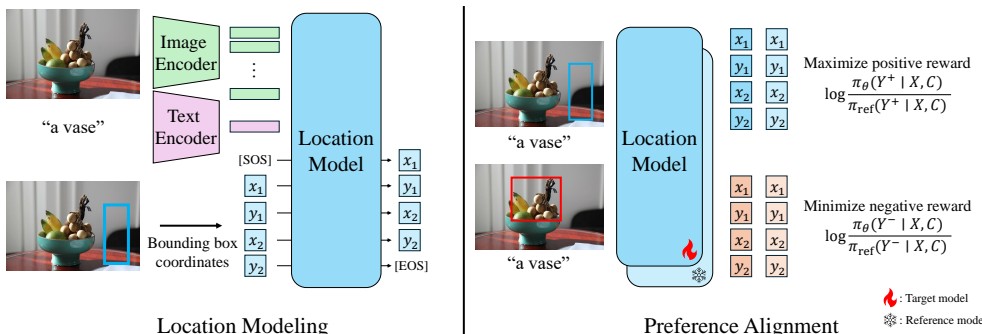

Figure 2: Training scheme during pretraining (left) and direct preference optimization (right).

$Y_i$ is represented as a bounding box with four components $\left[b_1^i, b_2^i, b_3^i, b_4^i\right] = [x_1, y_1, x_2, y_2]$, representing the coordinates of the top-left and bottom-right corners. Thus, given a dataset $\mathcal{D}$ which provides pairs of images $X$ and plausible locations $Y$ for object category $C$, we train the model using a negative log-likelihood objective:

$$\mathcal{L}_{\text{train}} = -\mathbb{E}_{(X,Y,C)\sim\mathcal{D}} \sum_{Y_i \in Y} \left[ \sum_{k=1}^{4} \log P(b_k^i \mid b_{<k}^i, X, C) \right]. \tag{2}$$

where each bounding box coordinate $b_k^i$ is sequentially predicted, conditioned on previous coordinates $b_{<k}^i$, the image $X$, and the object class $C$.

It is important to note that at training time, we model multiple bounding boxes independently (Equation 1) by predicting a *single bounding box* (*i.e.*, four coordinates) for a given input. In other words, we sample a single location $Y_i$ from $Y$ during training. This choice allows us to avoid issues related to the ordering of multiple plausible bounding boxes and arbitrary sequence lengths due to the sparsity of the annotations. During inference, we are still able to produce multiple locations by independently sampling multiple times.

The model architecture and the training procedure are illustrated in Figure 2. We tokenize images and the class embeddings using a pre-trained Vision Transformer (ViT) (Dosovitskiy, 2021) and a CLIP encoder (Radford et al., 2021). We encode bounding box coordinates by quantizing them to a grid with equally spaced bins of 1 pixel wide, (*i.e.*, 512 location tokens for $512 \times 512$ images). The image tokens and the target class token are prepended to the sequence, and our location model is trained to predict the probability of each coordinate in an autoregressive manner.

## 3.2 LEVERAGING NEGATIVE LABELS VIA DIRECT PREFERENCE OPTIMIZATION

While the training objective in Equation 2 allows training the location model on sparse positive annotations, training solely on positive feedback can lead to predictions in implausible locations. Incorporating negative annotations, when available, into the training objective can be beneficial for refining the model, encouraging it to assign lower likelihoods to undesirable locations and thereby improving overall accuracy.

Our generative formulation allows us to use any negative labels in the dataset as well, so that the model can learn to avoid predicting bounding boxes for implausible locations. Specifically, we treat the positive and negative labels as a preference dataset, where positive locations are implicitly preferred over negative ones, even though annotators were not explicitly asked to rank them. Using this preference structure, we fine-tune the model with direct preference optimization (DPO) (Rafailov et al., 2024), penalizing high logits assigned to negative labels. We repeat the training objective below and refer the reader to Rafailov et al. (2024) Eq. 1-6 for a thorough derivation.

Given a target location model $\pi_\theta$ (*i.e.* the model currently being finetuned by DPO), a reference location model $\pi_{\text{ref}}$ (*i.e.* a frozen model trained by Equation 2 only), and a preference dataset $\mathcal{D}_{\text{DPO}}$ where locations $Y^+$ are preferred over $Y^-$, we can derive the likelihood of $Y^+$ being preferred over $Y^-$, based on how well the target location model $\pi_\theta$ predicts each location relative to $\pi_{\text{ref}}$, following

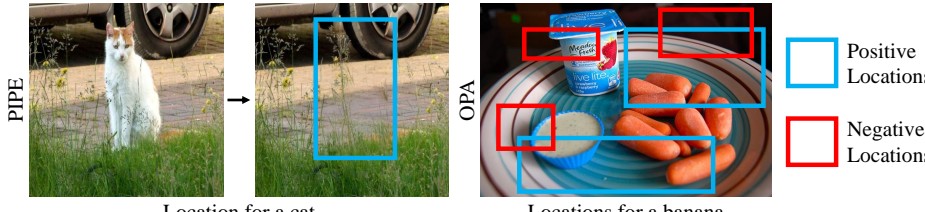

Figure 3: Annotation format for the PIPE dataset (left) and OPA dataset (right). The PIPE dataset has one groundtruth location per image, whereas OPA provides multiple positive and negative locations.

the Bradley-Terry model (Bradley & Terry, 1952):

$$P(Y^+ \succ Y^- \mid X, C) = \left(1 + \exp\left(\beta \log \frac{\pi_\theta(Y^- \mid X, C)}{\pi_{\text{ref}}(Y^- \mid X, C)} - \beta \log \frac{\pi_\theta(Y^+ \mid X, C)}{\pi_{\text{ref}}(Y^+ \mid X, C)}\right)\right)^{-1}. \quad (3)$$

Here, $\pi_\theta$ and $\pi_{\text{ref}}$ output the logits for the target model and the reference model, respectively, and $\beta$ is a hyperparameter. We can maximize the preference of $Y^+$ locations by initializing the target and reference models from a pre-trained location model, and then optimizing the target model using a negative log-likelihood objective:

$$\mathcal{L}_{\text{DPO}} = -\mathbb{E}_{(Y^+, Y^-, X, C) \sim \mathcal{D}_{\text{DPO}}} \left[\log \sigma\left(\beta \log \frac{\pi_\theta(Y^+ \mid X, C)}{\pi_{\text{ref}}(Y^+ \mid X, C)} - \beta \log \frac{\pi_\theta(Y^- \mid X, C)}{\pi_{\text{ref}}(Y^- \mid X, C)}\right)\right]. \quad (4)$$

In this way, we are able to leverage any available negative labels in the object placement dataset, and thereby improve the accuracy of the location model.

## 4 EXPERIMENTS

### 4.1 DATASETS AND ARCHITECTURE

**PIPE Dataset.** The PIPE dataset (Wasserman et al., 2024) was created by removing objects from object detection datasets (Lin et al., 2014; Kuznetsova et al., 2020; Gupta et al., 2019) by inpainting. This process results in pairs of images, one including the object and the other without it. To train our location model we need positive bounding box locations, that we derive for the missing object by thresholding the pixel-wise difference between the two images. An example is reported in Figure 3 (left). While the dataset offers a large number of 888,000 samples, many images contain inpainting artifacts, potentially introducing noise in the bounding box extraction process.

**OPA Dataset.** The OPA dataset (Liu et al., 2021) was created by asking human annotators to judge the plausibility of object placement locations for a subset of COCO images (Lin et al., 2014). This dataset includes on average 41.5 annotations per image, and can be used to encourage diversity in location model predictions (see Figure 3 right for an example). The train set includes 1022 images, and the test set include 130 images. As OPA provides negative labels, we also use OPA as the preference dataset for DPO training. Note that instruction-tuned editing models cannot leverage this location dataset, as they require fully rendered images for training.

**Implementation details.** We use a small GPT-2 (Radford et al., 2019) architecture as our autoregressive location model. For tokenizing the image and the object class, we use a ViT (Dosovitskiy, 2021) model pre-trained on ImageNet-21K (Ridnik et al., 2021) as our image encoder and the CLIP text encoder (Radford et al., 2021) as our object class encoder. We pre-train on PIPE for 30K iterations and finetune on OPA for 3K iterations. For batch size 64, the model can be trained on a single Nvidia V100 GPU. We quantize each box coordinate into one of 512 bins (*i.e.*, one bin per pixel). For DPO training, we train using the OPA dataset for 4K iterations. Please refer to Section D of the Appendix for further details.

We evaluate the performance of our generative location model in an object insertion pipeline in Section 4.2, where we rely on PowerPaint (Zhuang et al., 2023) as the inpainting approach. Specifically, we utilize the V2 version[1], built on top of BrushNet (Ju et al., 2024). We remark that our pipeline has the flexibility to incorporate any inpainting method for performing localized edits (see later in Section 4.4).

---

[1]https://github.com/open-mmlab/PowerPaint

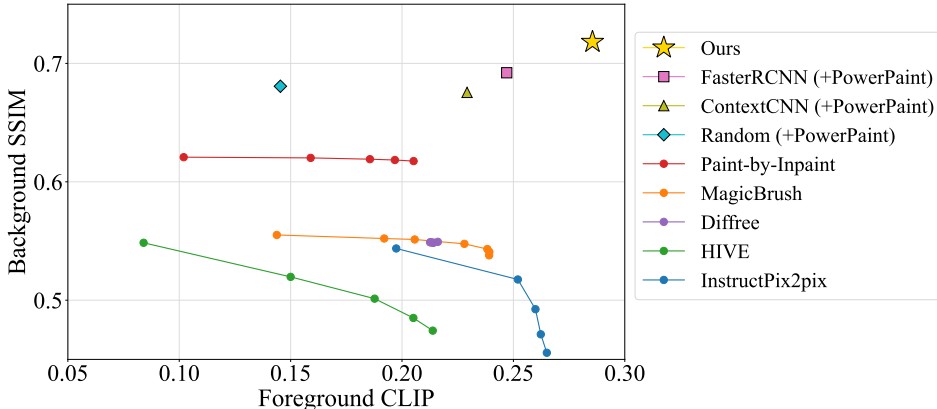

Figure 4: Quantitative evaluation on the OPA dataset, higher and to the right is best. For instruction-tuned approaches, each dot represents a different guidance scale ranging from 2 to 10. For other methods, guidance scale has a negligible effect, hence we show a single point.

## 4.2 OBJECT INSERTION

**Baselines.** We compare our object insertion pipeline against recent instruction-tuned image editing models, which represent some of the most advanced techniques for object insertion to date. Specifically, we evaluate against general-purpose image editing models InstructPix2Pix (Brooks et al., 2023), HIVE (Zhang et al., 2024b), and MagicBrush (Zhang et al., 2024a), as well as object insertion-specific models, Paint-by-Inpaint (Wasserman et al., 2024) and Diffree (Zhao et al., 2024).

Moreover, we experiment with three additional explicit location models that we also pair with the PowerPaint inpainter. First, we test random locations as a simple baseline. Then, we train ContextCNN (Dvornik et al., 2018), a classifier designed to assess masked regions of an image, determining their suitability for object placement. To be able to use it with the OPA class space, we retrain the model on COCO (whose classes comprehend all the ones in OPA). Furthermore, we train a Faster-RCNN object detector (Ren et al., 2015) using positive OPA annotations to serve as a high-performing discriminative model.

**Evaluation Metrics.** Similar to existing image editing benchmarks (Wasserman et al., 2024), we evaluate the success of object insertion by considering both how well the original scene is preserved and how accurately the target object is inserted. Using an object detector (Zhu et al., 2021) we separate the background, which ideally should remain untouched, and the foreground, where the newly inserted object appears.

To measure background preservation, we compute the Structural Similarity Index Measure (SSIM) (Wang et al., 2004) on the background region. To assess the accuracy of the object, we measure the CLIP similarity (Radford et al., 2021) between the cropped object and the text "*an {object class}*". If no new object is detected, we assign a CLIP score of 0 to reflect the failure to properly insert the object.

For the PIPE dataset, we also evaluate the diversity of edits by calculating the average LPIPS distance (Zhang et al., 2018) across 10 different edits per background-object pair. High LPIPS indicates that a model can generate diverse results from the same instruction, highlighting approaches that are limited to producing a single edit.

**Results.** We compare instruction-finetuned image editing models, location models paired with strong inpainting models, and our approach in Figure 4. Our approach substantially outperforms all baselines by leveraging a dedicated location model for inpainting, which effectively reduces background distortions while maintaining high-quality object generation.

By design, localized inpainting is an effective strategy for preserving the background, but it is notable that even a random location model outperforms the best instruction-finetuned method on background SSIM. Since the key difference between our approach and other location modeling baselines lies in the plausibility of the predicted locations, the figure suggests that an accurate placement directly impacts the quality of inpainting results.

Table 1: Evaluation on the PIPE benchmark.

|  | BG SSIM | FG CLIP | LPIPS |
|---|---|---|---|
| InstructPix2Pix | 0.5679 | 0.2670 | **0.3865** |
| MagicBrush | 0.6118 | 0.2579 | 0.0986 |
| HIVE | 0.5635 | 0.2047 | 0.1886 |
| Diffree | 0.6170 | 0.2517 | 0.1210 |
| Paint-by-Inpaint | 0.7281 | 0.2672 | 0.0700 |
| Ours | **0.8075** | **0.2774** | 0.1824 |

Figure 5: User study results on edited images (OPA). Left, blue: ours preferred. Right, red: baseline preferred. Middle, grey: no preference.

The instruction-tuned models often face a trade-off: they either preserve the background well but fail to generate the object convincingly, or they successfully generate the object but significantly alter the surrounding scene. This inconsistency may stem from the fact that these models address both the placement and generation tasks simultaneously, leading to suboptimal object locations. In contrast, using a dedicated location model allows one model to focus on spatial reasoning, while the inpainting method concentrates on rendering realistic objects.

We further evaluate our approach on the PIPE test set (Wasserman et al., 2024), which contains images with specific objects removed. Using the same metrics reported in Table 1, our results are consistent with those observed in the OPA test set. Notably, Paint-by-Inpaint (Wasserman et al., 2024) lacks diversity, frequently generating identical outputs, as shown in Figure 8. This is likely due to training on datasets created by object removal through inpainting. Moreover, we appreciate how InstructPix2Pix and HIVE achieve very diverse edits. However, we notice that this result is typically achieved by generating entirely new images, rather than editing the input scene, as also testified by their low background SSIM scores.

**User Study.** To further validate that our model's insertions are favored by human observers, we perform a user user study comparing our method to the four strongest baselines. We show pairs of edited OPA images and ask users to indicate which of the two is the better edit, for 46 participants, each of which ranks 40 image pairs. For further details, see Appendix Section F. Results are shown in Figure 5. Participants preferred edits generated by our approach over those from baseline approaches, indicating that our metrics agree with human preference, and that better edit quality can be achieved through precise location modeling. Additional qualitative examples can be found in Figure 7, Figure 8, and Appendix Section G.

### 4.3 LOCATION MODELING

**Baselines.** We compare our generative location model against two discriminative approaches that classify locations as plausible or implausible, namely ContextCNN (Dvornik et al., 2018) and Faster-RCNN (Ren et al., 2015), as described in Section 4.2. Additionally, we compare to two object placement baselines (Zhou et al., 2022; Niu et al., 2022) that perform placement of specific object segments rather than generic class labels. We use the official implementations relying on foreground segments available within the OPA dataset. Note that a direct comparison to these two methods is challenging, as our method does not use the object segment, and it is hard to say whether access to one makes the task harder or easier. Still, we include these baselines for completeness.

**Evaluation Metrics.** The OPA test set provides plausible (positive) and implausible (negative) locations for objects given an image. However, due to the sparse nature of these annotations, it is impossible to sample locations until every ground-truth bounding box is matched with a prediction. Therefore, we evaluate location models based on a "hit rate" metric, which compares the rate of each predicted box being a plausible or implausible location. Specifically, we measure the True Positive Rate (TPR) and False Positive Rate (FPR) for a given set of predicted locations. Given $K$ predicted locations, we match them to the ground-truth labels using the Hungarian algorithm (Kuhn, 1955), where the cost function is the inverse of the intersection over union (IoU).

True positive predictions are defined as predictions assigned to positive labels with an IoU above 0.7, while false positive predictions are those assigned to negative labels under the same IoU threshold. Any positive or negative ground-truth locations that are not matched are counted as false negatives

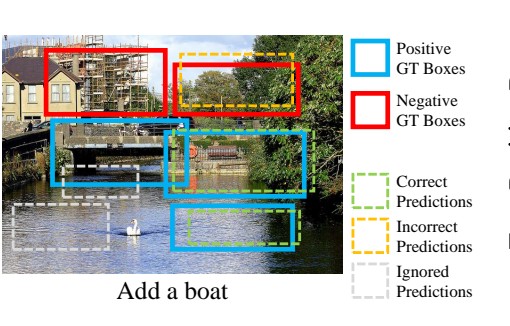
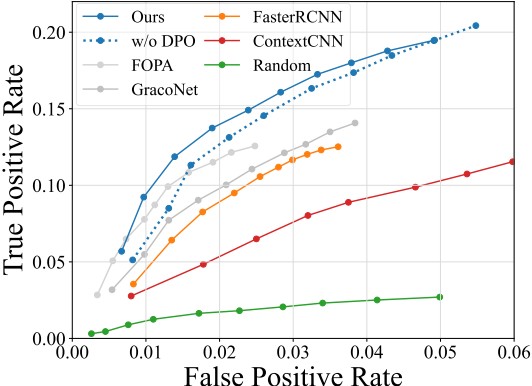

Figure 6: (Left) Example evaluation scenario. Predicted boxes are counted only if a positive or negative ground-truth box meets an IoU above the threshold. (Right) TPR-FPR curves. Each line is constructed by sampling $\{10, 20, \ldots, 100\}$ locations. Top-left is better.

and true negatives. Predictions that do not correspond to any labeled locations are ignored, as their true labels cannot be determined. We refer the reader to Figure 6 (left) for an example of such assignments. TPR and FPR are then computed using standard definitions. Intuitively, TPR represents the rate of a predicted location being a correct (positive) location, and FPR represents the rate of a predicted location being an incorrect (negative) location.

**Results.** We plot the TPR and FPR for different number of sampled locations $K = 10, 20, \ldots, 100$ in Figure 6 (right). Our generative location model consistently achieves a higher TPR at the same FPR, appearing in the top-left region of the plot. In contrast, discriminative baselines (Dvornik et al., 2018; Ren et al., 2015) fail to reach a high TPR, even after predicting 100 locations, possibly due to the penalization of unlabeled positive locations during training. Our generative approach avoids any assumption about unlabeled locations, allowing it to achieve higher accuracy in identifying plausible ones. This finding suggests the effectiveness of generative modeling in scenarios where the sparsity in annotations hinders training a discriminative model.

### 4.4 ABLATION STUDY

**DPO Training.** Training a generative location model exclusively on positive locations already demonstrates strong performance, as shown in Figure 6 (right). However, incorporating negative labels further enhances accuracy by explicitly guiding the model on where *not* to predict object locations. Notably, unlike existing location modeling techniques, which assume non-labeled locations as negative even when negative labels are present, our approach leverages only the negative labels provided by annotators. This ensures more precise predictions, as it avoids the potential inaccuracies introduced by assuming non-labeled locations are negative.

**Alternative Inpainting Methods.** Our location model is not tied to a single inpainting method and can incorporate various inpainting techniques (Xie et al., 2023; Ju et al., 2024; Cao et al., 2024). To illustrate this, we use the same predicted locations to render objects with the inpainting model of GLIGEN (Li et al., 2023). The resulting images rendered achieve a background SSIM of 0.6511 and foreground CLIP of 0.2833, compared to a performance of 0.7184 background SSIM, 0.2849 foreground CLIP, when using PowerPaint. Obtaining similar foreground CLIP score indicates that inpainting is often successful even with the older GLIGEN model. We therefore expect that our approach benefits from future advancements in inpainting techniques as well.

### 5 DISCUSSION

**Inference Cost.** Our location model introduces a minimal overhead to the overall object insertion process. To measure this, we compare the inference time of the location model relative to the time required for rendering the image (*i.e.*, inpainting). On an average across 100 runs, our model takes 0.03 seconds to sample a single location on a Nvidia Tesla V100 GPU, a minor addition compared to the 7.10 seconds needed to render an image using a 50-step diffusion reverse process

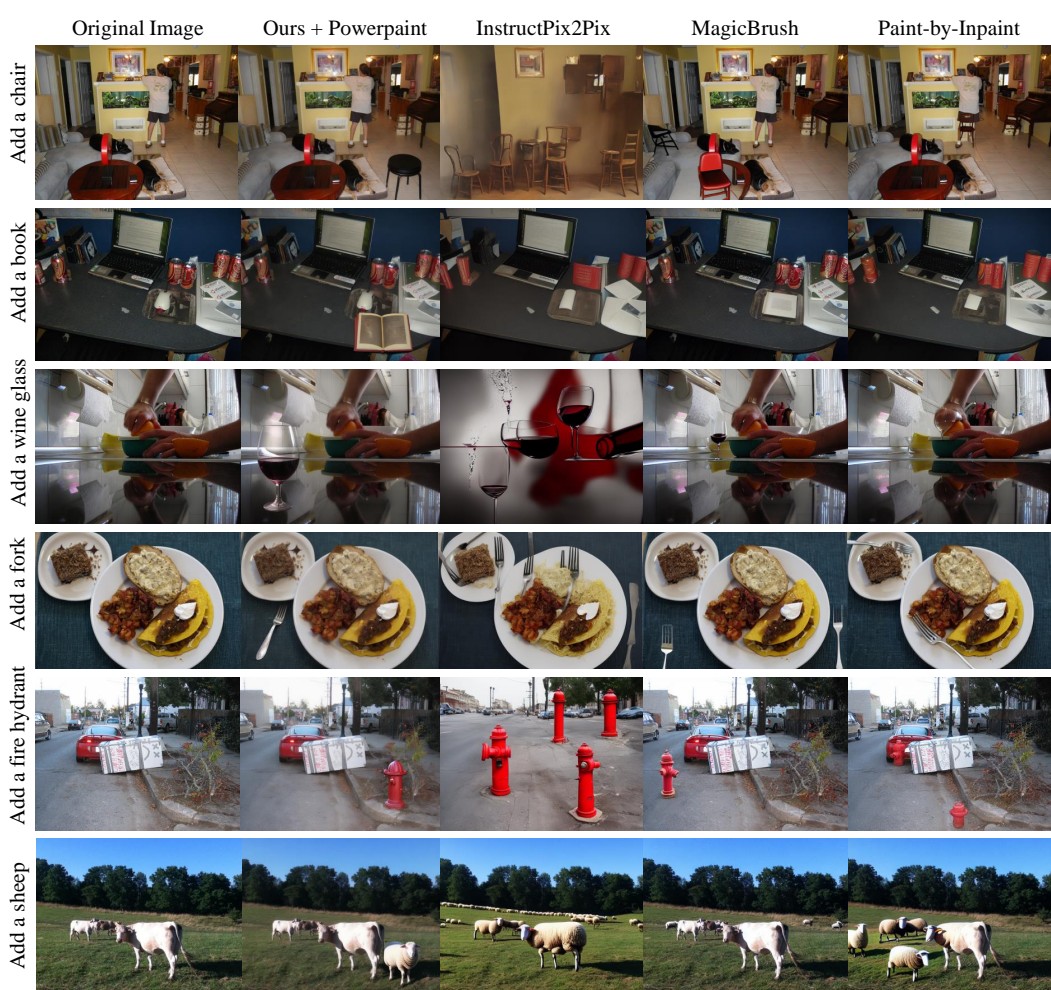

Figure 7: Comparison between our method + powerpaint, and instruction-guided image editing models on the OPA dataset. Best viewed electronically.

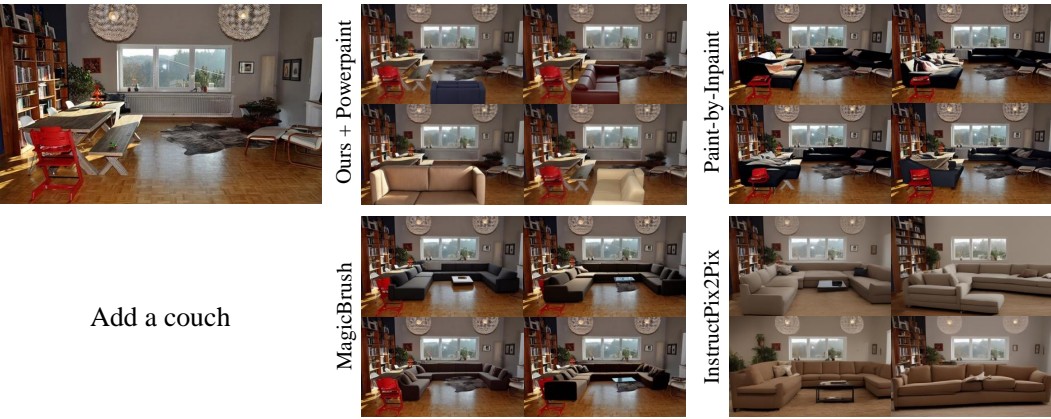

Figure 8: Comparison between our method + powerpaint, and instruction-guided image editing models on the PIPE dataset. Additional results are available in Section G. Best viewed electronically.

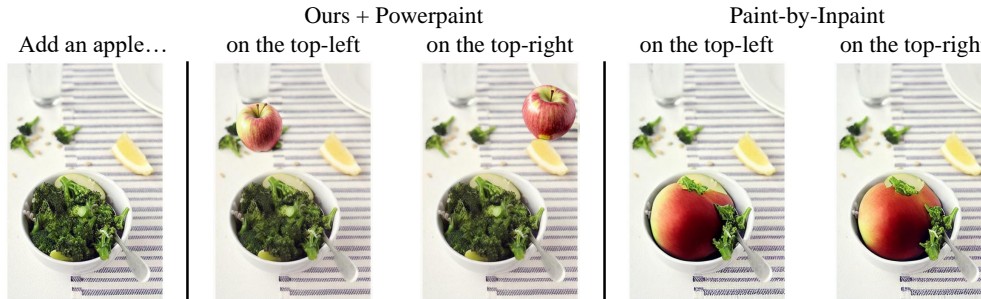

Figure 9: Controlled location sampling for positional instructions, achieved by restricting the allowed sampling region. Instruction-tuned models often fail to follow positional instructions.

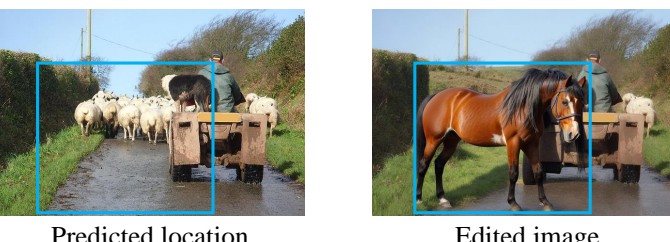

Figure 10: Example failure case observed when inpainting large areas, for instruction "add a horse".

for a StableDiffusion v2.1 checkpoint (Rombach et al., 2021). Paying a small upfront cost for identifying a plausible location leads to a significant improvement in the quality of object insertion.

**Controlled Location Sampling.** A location model not only automates the process of determining where to insert objects but also allows users to specify locations that meet particular requirements. Current instruction-tuned image editing models struggle with positional instructions, such as "Add an apple to the top-left of the image." In contrast, our location model can handle these requests by constraining model outputs to specific areas, such as by sampling only the top-left coordinates. As illustrated in Figure 9, by factorizing the object insertion process into a location modeling step and an inpainting step, our approach offers increased control over object placement.

**Limitations.** Although localized object insertion helps minimize distortions in the background, predicted bounding box locations may sometimes occlude more of the background than necessary. As seen in Figure 10, large bounding boxes may lead to unwanted changes in the scene, such as changes to the background or inadvertently occluding foreground objects. It may also result in unrealistic background effects, such as missing shadows or reflections. These issues could potentially be mitigated by predicting fine inpainting masks within the bounding box regions or by developing inpainting techniques tailored specifically for object insertion, which we leave as future work.

## 6 CONCLUSION

In this paper, we present a location model that identifies plausible locations for objects to be inserted within an image. By taking a generative approach, we are able to work with sparsely annotated location datasets. We also show that it is beneficial to use negative labels via direct preference optimization. Our experiments suggest that in the task of object insertion, separating the problems of *where* to place an object and *what* to place in the given location, is currently much more reliable than instruction-tuned insertion approaches.

More generally, we believe that building spatial awareness is a key factor for building reliable models interacting with the real world. This is true whether a model operates in the image editing setting, as in this work, or in more complex domains, such as robotics or virtual reality. Our work shows that a model is able to learn such awareness from example positive and negative annotations. Although we focus on locations for 2D bounding boxes conditioned on images, we hope to see similar models scale to handle other scene representations (*e.g.*, depth or semantic layouts) and precise locations (*e.g.*, 3D bounding boxes or object masks) in the future.

**Broader Impact Statement** Spatial awareness is crucial for real-world applications where precise and context-aware object placement is necessary. Improving the reliability of object insertion methods can benefit numerous fields, such as augmented reality, robotics, and generative data augmentation, by enabling more realistic and practical scene manipulations. However, image editing methods can also be misused, for example, to manipulate images by inserting individuals into scenes to damage their reputation. Our method, which specifies only the desired object class rather compositing a background image and a specific target object, is hopefully less suitable for such malicious use cases. Nevertheless, one should consider malicious use cases of image editing methods before deployment, especially if used in real world or safety-critical applications.

**Reproducibility Statement** We provide comprehensive details of our architecture and data preprocessing steps in Section 4.1 and Section D. Our architectures are based on open-source implementation of GPT-2 (Radford et al., 2019), for example found at this github repo, and uses standard open source vision backbones to initialize the vision and text encoders. Furthermore, all datasets used in our experiments (Liu et al., 2021; Wasserman et al., 2024) are publicly available, facilitating the replication of our work by the community.

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

# APPENDIX

## A  DATASET STATISTICS AND PREPROCESSING

### A.1  PIPE DATASET

The PIPE dataset (Wasserman et al., 2024) was created by removing objects from object detection datasets (Lin et al., 2014; Kuznetsova et al., 2020; Gupta et al., 2019) using an inpainting model, resulting in over 600 object classes for insertion and more than 888,000 training pairs of images, showing scenes before and after object removal. To preprocess this dataset, we pair each background image with the original location of the removed object. The locations are identified by computing the pixel-wise difference between the before-and-after images, and extracting coordinates where the difference exceeds a certain threshold. Despite this thresholding, the resulting bounding boxes can be noisy, and the background images generated by the inpainting model often contain artifacts. Also, the PIPE dataset only provides a single positive location for an object for each background image and does not provide negative labels.

### A.2  OPA DATASET

The OPA dataset was created by manually annotating samples from the COCO dataset (Lin et al., 2014), resulting in 47 object categories for object placement. The dataset is intended for the task of finding locations to copy-paste images of objects, and therefore includes background images, object images with transparent backgrounds, and labeled plausible/implausible locations for placing the objects. Since our focus is on object location modeling rather than the insertion of object images, we ignore the object images and restructure the dataset to include pairs of background images and their corresponding object locations. This restructuring yields 1,496 training samples and 184 test samples. While the number of images is relatively small, each sample contains around 40 annotations, making it a richly annotated dataset. In total, the OPA train set contains 21,376 positive labels and 40,698 negative labels.

Despite having an average 40 annotations for each sample, this accounts for fewer than 1% of the typical number of anchor boxes used in object detectors, leaving most locations unlabeled. Furthermore, as illustrated in Figure 11, not only are the number of bounding boxes across samples highly imbalanced, but the distribution of positive and negative labels for each sample is extremely inconsistent. This sparsity and imbalance make it extremely challenging to train discriminative models that classify locations as plausible or implausible. Our generative approach bypasses this issue by modeling the distribution of plausible locations, using negative labels only for DPO.

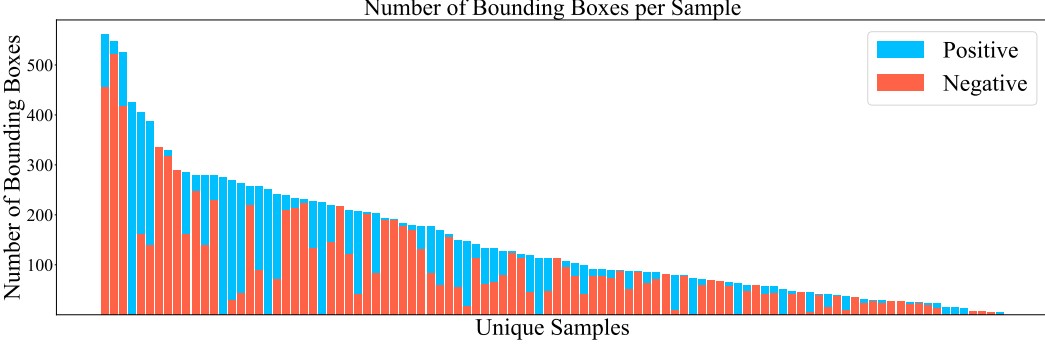

Figure 11: Distribution of positive and negative bounding boxes in the OPA dataset. We randomly select 100 samples (images and their annotations) from the training set for visualization.

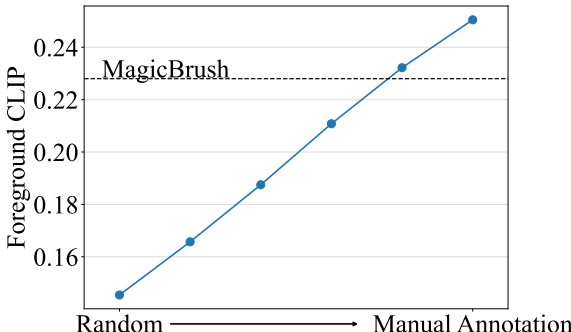

Figure 12: Inpainting quality with respect to the quality of the location. The success of inpainting approaches for object insertion is highly dependant on the quality of the location.

## B    DOES LOCATION MATTER?

To demonstrate that the quality of locations directly influences the quality of generated objects, we present a proof-of-concept experiment in Section 12. For each image in the OPA dataset, we uniformly sample 10 random locations and then interpolate between these random locations and manually annotated ones, performing inpainting at six distinct interpolation points. To evaluate the success of inpainting, we measure the foreground CLIP similarity (Radford et al., 2021) between the cropped object and the text "*an {object class}*". For reference, we also include the performance of MagicBrush, which inserts objects without explicit location modeling. As the inpainting locations become more precise, the fidelity of generated objects increases, underscoring the importance of accurate locations. Inpainting in incorrect locations often results in failed insertions, motivating the development of a dedicated location model to provide spatial awareness for inpainting models.

## C    ALTERNATIVE APPROACHES FOR LOCATION MODELING

Vision-language models (VLMs) can also be used to predict plausible locations for object placement based on image inputs. Although these models can generate bounding box coordinates as part of their responses, we find that they are largely ineffective at predicting meaningful locations, with their bounding box outputs being comparable to random guesses. Specifically, we use LLaVA-13B (Liu et al., 2024) using the following prompt:

```
USER: <image>If a new {object_name} would appear in this scene,
what would be the coordinates of the {num_samples} different
plausible locations? "Answer in JSON format
    {
        plausible location 1\": [x1, y1, x2, y2],
        plausible location 2\": [x1, y1, x2, y2], ...,
        plausible location {num_samples}: [x1, y1, x2, y2]
    }.
Output must only include the JSON format and no other text.
ASSISTANT: In this scene, {num_samples} most plausible locations
of a newly inserted {object_name} are:
```

We also compare against object placement approaches (Zhou et al., 2022; Niu et al., 2022) that predict locations based on both the background image and a tightly masked image of the object. Since these models are provided with the exact aspect ratio of the object, they only need to predict the location and scale of the bounding box. As previously mentioned in Section A, the OPA dataset includes object images, allowing us to measure the TPR and FPR on the same test set. Despite having the advantage of having provided with ground-truth aspect ratios, these models are outperformed by our location model, which demonstrates superior performance even without access to object images. A full comparison of these location models are plotted in Figure 13.

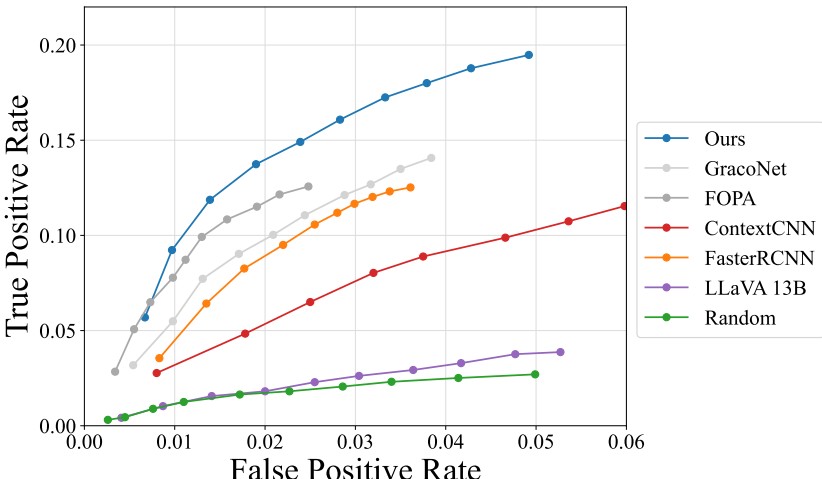

Figure 13: TPR-FPR curves compared with object placement approaches and LLaVA.

# D    ARCHITECTURE AND IMPLEMENTATION DETAILS

## D.1    ARCHITECTURE OF THE LOCATION MODEL

Our location model is based on a GPT-2 small architecture (Radford et al., 2019), consisting of 12 layers of Transformer blocks (Vaswani et al., 2017). For image and object class encoding, we use a pre-trained ViT-B model (Wightman, 2019; Dosovitskiy, 2021) for the image encoder and a ViT-B CLIP text encoder (Ilharco et al., 2021; Radford et al., 2021) for the object class. The images are converted into 196 embeddings, while the text is transformed into a single embedding, creating a sequence length of 197 when combined. These 197 embeddings are prepended to our location model before predicting the coordinates. To quantize the coordinates, we use 512 bins for both height and width, resulting in a vocabulary size of 514, including the start-of-sequence (SOS) and end-of-sequence (EOS) tokens.

The location model, including the ViT-B backbone, comprises a total of 411 million parameters, and loading the weights in float16 precision requires just 2.05 GB of VRAM. Running the model with batch size 1 in float16 precision requires an additional 1.93 GB of VRAM for the activations, meaning the model can easily be run on consumer-grade hardware. On a Nvidia Tesla V100 GPU, inference runs in 0.03 seconds. For editing images, we first sample locations, unload the model from the GPU, and perform inpainting, ensuring no additional memory overhead beyond what the inpainting model requires.

In comparison, the PowerPaint inpainting model has three components: a VAE with 83.7 million parameters, a text encoder with 123.1 million parameters, and a UNet with 859.5 million parameters. Although the location model is nearly half the size of the UNet in terms of parameters, it is used only once in advance (similar to the VAE and text encoder), whereas the UNet is used at every reverse step. On the same GPU, the it takes 7.10 seconds to perform 50 reverse steps, meaning the inference overhead of the location model is 0.4%, an acceptable cost for the improvement in generation quality.

## D.2    TRAINING DETAILS

We train the model using the Adam optimizer (Kingma, 2015) for training the location model and Stochastic Gradient Descent (SGD) for DPO training. The model is trained on the PIPE dataset for 30,000 iterations with a learning rate of 1e-4, incorporating a linear warmup over the first 1,000 steps. Subsequently, we fine-tune the model on the OPA dataset for 3,600 steps and perform DPO for an additional 4,600 steps. For batch size 64, the model can be trained on a single Nvidia V100 GPU.

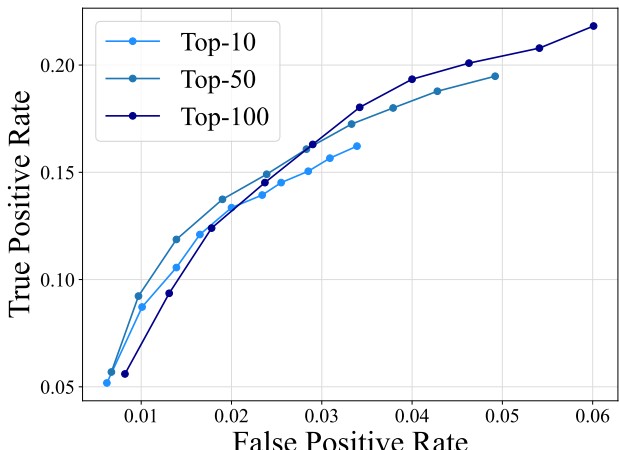

Figure 14: Location modeling performance for different top-$k$ parameters used during sampling. Higher numbers of $k$ leads to diverse predictions, often at the cost of accuracy.

### D.3 SAMPLING FROM THE LOCATIONS

Our autoregressive layout model can follow sampling techniques similar to those used for text generation using Large Language Models (LLMs) (Radford et al., 2019). We sample among the top-$k$ probabilities scaled with a temperature of $1.0$. As illustrated in Figure 14, we find that the higher values of $k$ promotes diversity and thus achieves a higher True Positive Rate (TPR), but also increases the False Positive Rate (FPR). For the main experiments, we use $k = 50$ during sampling.

## E DIVERSITY OF THE SAMPLED LOCATION

An essential aspect of location models is their ability to identify diverse plausible locations, as multiple potential placements often exist for a given object. To quantify this diversity, we apply Non-maximum Suppression (NMS) to the predicted locations and count the remaining bounding boxes. Specifically, we perform a standard NMS with a threshold of 0.7 on a set of 100 predicted locations for the same object in the same scene. We observe that 82.5 bounding boxes remain after performing NMS, which highlights the diversity of our location model.

## F HUMAN EVALUATION

We use four baselines in our user study: three instruction-finetuned models, and the strongest location modeling baseline (*i.e.*, Faster-RCNN trained on OPA + Powerpaint).

Participants were presented with the original image, an instruction (*"Add a {object class}"*), and two edited images: one generated by our approach and the other by an instruction-tuned editing model or location modeling baseline. For each image, we randomly swap which method is shown on the left or right side. Each participant was asked to evaluate which of the two edited images better adhered to the editing instruction and maintained the overall coherence of the scene. In total, we collected 1,840 responses from 46 participants, with each individual comparing $4 \times 10$ pairs of randomly selected samples.

## G QUALITATIVE RESULTS

We provide additional image editing results from the PIPE dataset. In our observations, instruction-tuned models often lack diversity in their final edits when they successfully insert objects. When these models fail to insert objects, they either leave the image unchanged or modify too much of the scene, which are both regarded as a failure to insert objects.

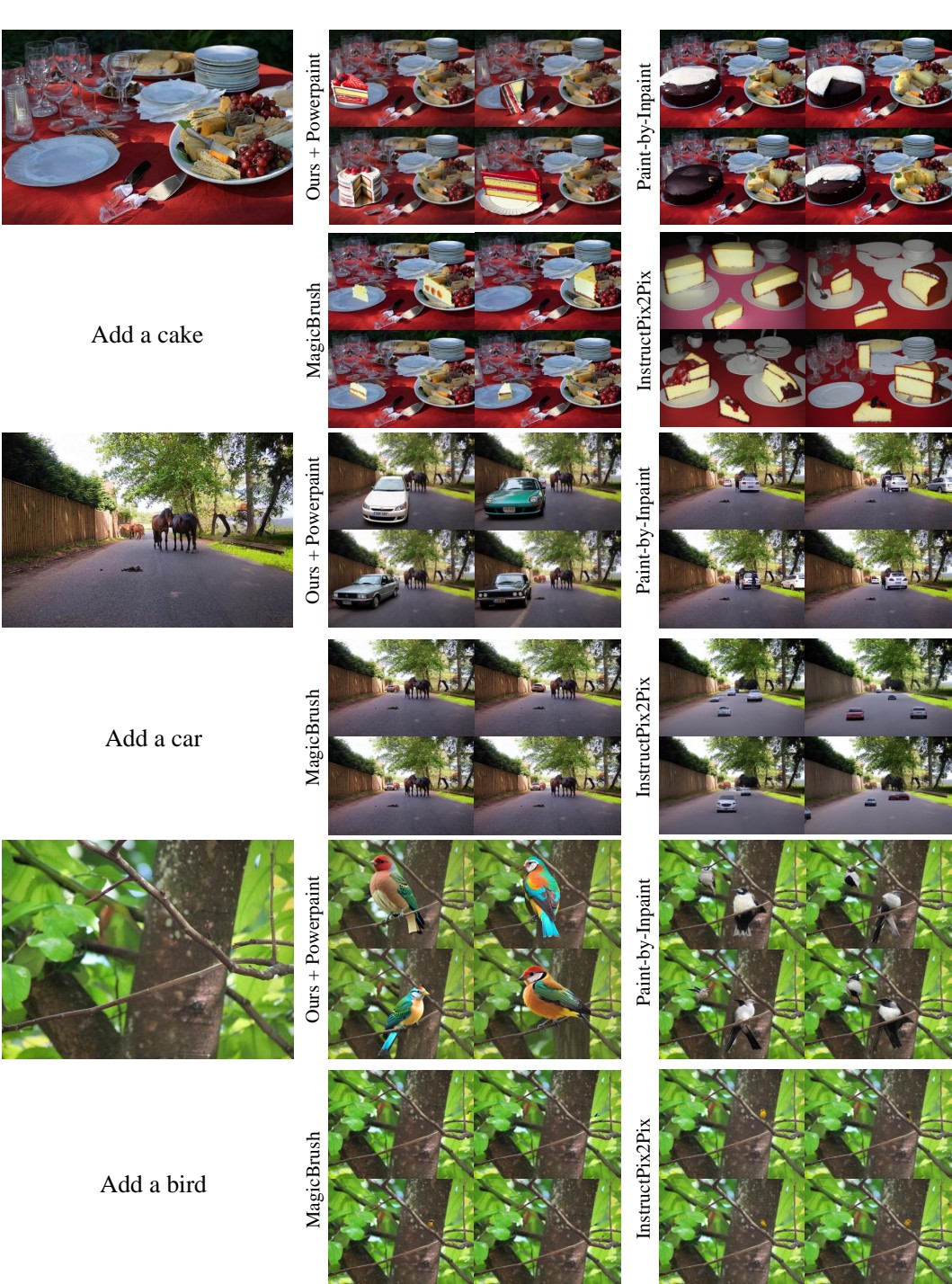

Figure 15: Additional samples of object insertion results in the PIPE test set.

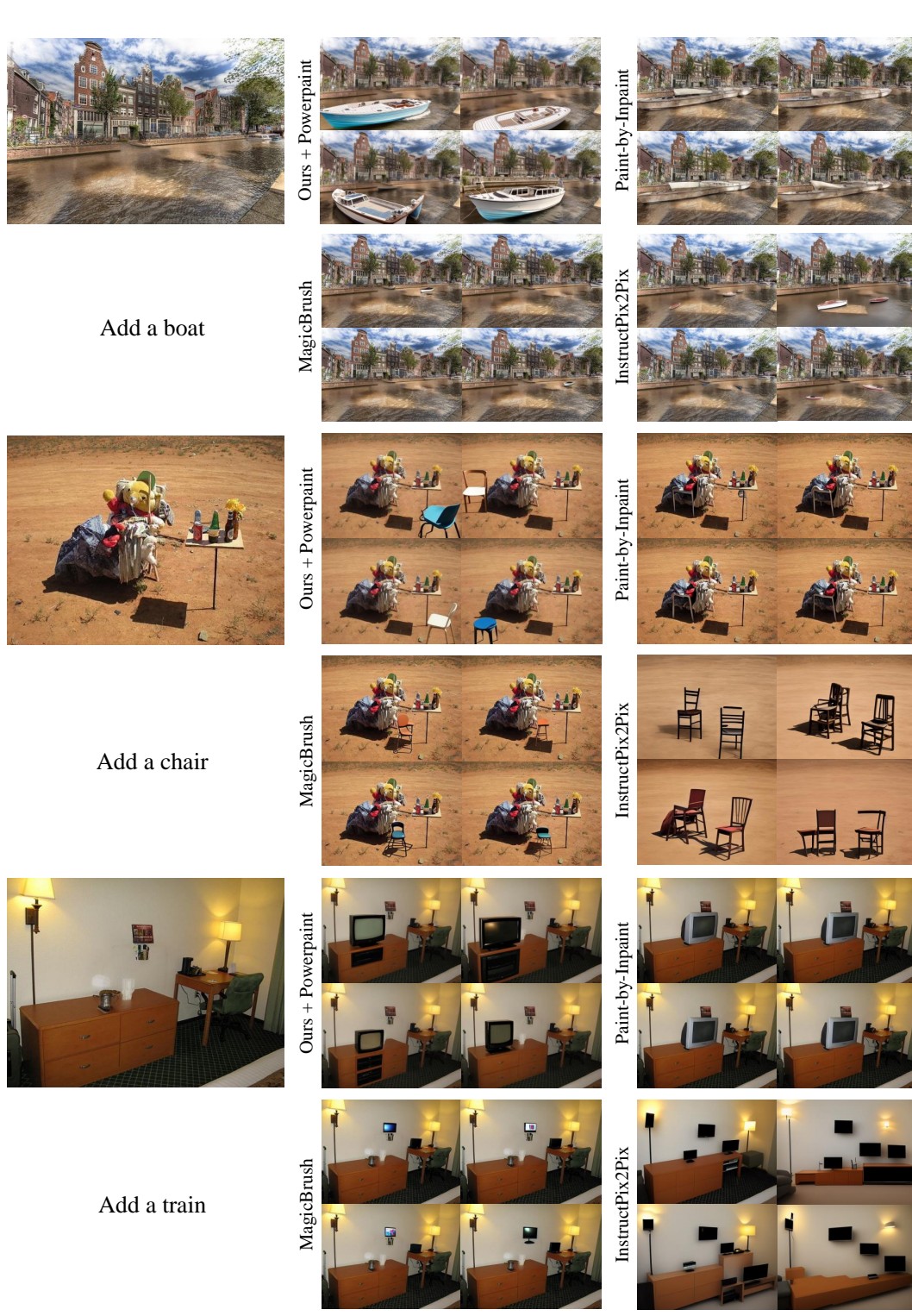

Figure 16: Additional samples of object insertion results in the PIPE test set.

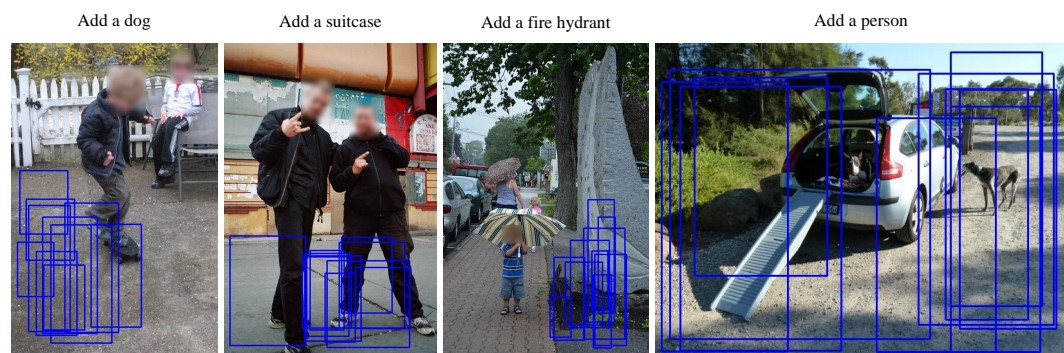

Figure 17: Bounding box proposals for various images and object categories from OPA.

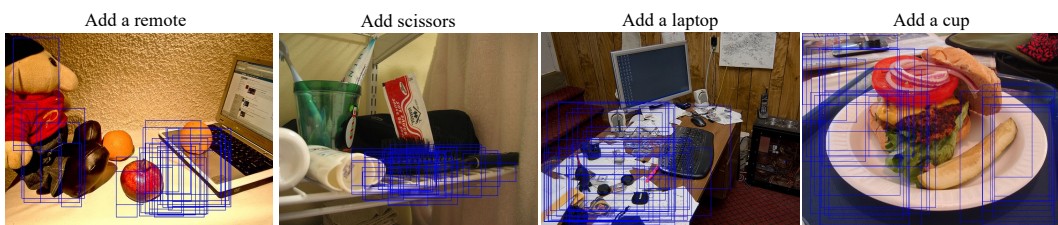

Figure 18: Bounding box proposals for various images and object categories from OPA, for cluttered scenes. Placement is more challenging if there is less empty space in the scene.

We also report additional visuals showing bounding box proposals for given images in Figure 17 and Figure 18. These figure highlights the diversity of the sampled locations from a location model, and show the behavior of the location model in cluttered scenes, where placement is more challenging.

Additionally, we show qualitative samples highlighting generalization capability in Figure 19. As the class is encoded using a CLIP text encoder, we can condition on class labels that are not present in the data and expect a reasonable output if the corresponding CLIP embedding is sufficiently close to existing categories. We show that we can add another instance of an already present object without our bounding box prediction coinciding with the existing instance, and that we can add instances from other classes in OPA and out-of-domain classes.

Finally, we show some additional failure cases of the location model in Figure 20. These can occur when the object category does not well in the scene, for example adding an airplane to a scene without much visible sky. As the inpainting model is not aware of existing objects, this can lead to occluded objects being removed from the scene.

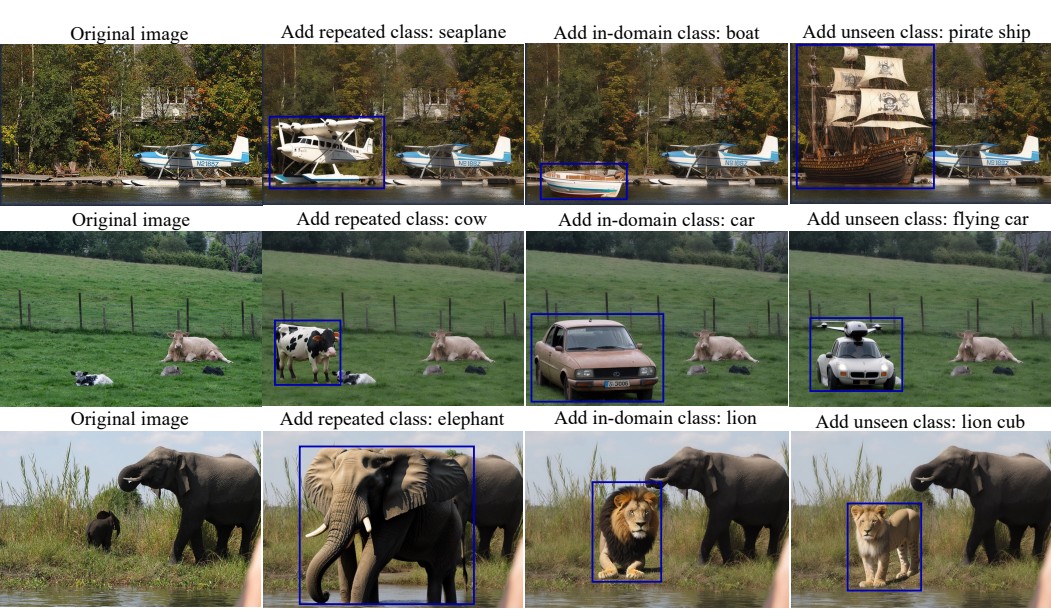

Figure 19: Qualitative examples showcasing generalization ability. We ask our model to add one more instance of an already present class (repeated), an instance of a different class (in-domain), and an instance of a class that does not exist in the OPA dataset (unseen).

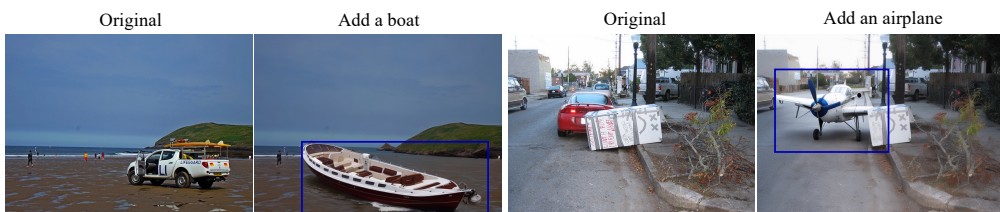

Figure 20: Failure cases of the location model leading to downstream issues during inpainting. When the space in which an object class can reasonably be placed is small, the predicted location may be incorrect, and the resulting inpainted object can inadvertently overwrite other objects.

