# OpenReview forum: "Generative Location Modeling for Spatially Aware Object Insertion"
_ICLR.cc/2025/Conference — Submitted to ICLR 2025_

### Official Review · Reviewer_QUkD · 2024-10-25

**Soundness:** 2
**Presentation:** 3
**Contribution:** 2
**Rating:** 5
**Confidence:** 4

**Summary:**

This work presents a two-stage approach for realistic object insertion, enhancing visual coherence by separating "where" and "what" tasks. The model first predicts plausible object positions using an autoregressive transformer, followed by inpainting for object insertion. Direct preference optimization (DPO) refines the location model, leveraging bounding box annotations to improve placement accuracy. Results show it outperforms instruction-based models in addressing unrealistic locations, background distortion, and scaling issues.

**Strengths:**

- The motivation is clear and the techniques sound reasonable.
- The writing and organization are easy to follow.
- Experimental results indicate that the proposed approach can achieve better performance.

**Weaknesses:**

1. The proposed approach lacks technical novelty, primarily combining existing location prediction and inpainting techniques. Improving the accuracy of bounding box predictions would naturally enhance the results of 2D/3D inpainting result.
2. 2D object insertion is a well-explored topic, with numerous works on 2D/3D insertion (including Gaussian Splatting and NeRF) presented at CVPR'24 and ECCV'24. The paper requires some fair and substantial comparison with baselines and other recent inpainting methods to justify its contributions.
3. This work relies heavily on the inpainting model to handle the object’s occlusions and interaction cases. The authors don’t mention how the model performs in highly cluttered or dense environments with multiple potential object placements.

**Questions:**

- The paper does not show how the model performs in complex environments where multiple potential object placements may exist, which is more promising for the community.
- I think the authors should do some study on decoupling instructions rather than simply predicting the positions.
- The paper could include more in-depth case studies and analysis, which would provide readers with a clearer understanding of the strengths and limitations of the proposed approach.

---

> ### Author Response · Authors · 2024-11-19
> **Initial response 11/19**
>
> ### Technical novelty
>
> We respectfully disagree about our proposal simply combining existing location prediction and inpainting techniques.
> *We do not rely on any pre-existing location model, but rather develop a novel generative solution for the problem.*
> Most modern approaches have predominantly used discriminative frameworks, resulting in techniques classifying given locations as either plausible or implausible.
> We believe our generative approach allows the model to learn directly from sparse annotations without any assumption about unlabeled regions, which is a limitation that might bias discriminative models.
> The choice of a generative model also enables the use of direct preference optimization, which we show to be effective in this context.
>
> Moreover, it is intuitive but not trivial that better bounding box locations enhance the result of the inpainting method.
> Where many image editing methods [1,2] focus on curating large scale datasets for training instruction-based models, very little attention is spent on improving the location of the edit explicitly.
> In Figure 12 in the Appendix, we show support for the intuition that working on edit location prediction is beneficial, and we hope that our work encourages more efforts in this direction.
>
> ### Addressing 2D and 3D object insertion approaches
>
> We would like to clarify our setting for object insertion, as it is a diversified field where the term is used in various contexts.
> Many existing works focus on inserting objects by either predicting the location *given object specifications* (e.g., meshes or images), or predicting the object appearance *given the location* (e.g. through a user-specified mask or box).
> In contrast, our work addresses scenarios where *both the location and the object appearance are unspecified*, and only the object category is provided.
> We consider use cases where it is challenging for users to prepare an object exemplar that seamlessly fits into the given scene.
> This setting makes location modeling particularly challenging, as the model must assess the plausibility of a location without any prior knowledge about the appearance of the added object itself.
> To the best of our knowledge, instruction-tuned image editing models are the most capable models that perform when both location and object specifications are not provided.
> We hope this clarification addresses the concern regarding other works that share the term "object insertion".
> If the reviewer has specific papers in mind that they believe are relevant to our setting, we would greatly appreciate it if they could share them with us.
>
> Furthermore, while text-based 3D object insertion methods, such as Instruct-NeRF2NeRF [3], also insert objects into scenes, they fall outside the scope of our work as they rely on additional scene information (e.g., videos or 3D information) for reconstruction.
> Nonetheless, a location model could still be valuable in such scenarios by providing bounding box locations, as recent findings such as InseRF [4], have demonstrated using user-provided boxes.
>
> ### Handling multiple locations
>
> We strongly agree with the reviewer that diversity in location predictions is a valuable trait for location models, particularly in complex real-world scenarios.
> We first wish to highlight that the OPA dataset features multiple plausible locations per image, averaging to 40 annotations per image (see Section A.2 of the appendix), capturing the natural diversity of plausible locations.
> Additionally, the OPA evaluation procedure in Section 4.3 considers the nature of these annotations and it rewards diversity: *a model can only achieve a high True Positive Rate and a low False Positive Rate on OPA if it generates diverse and accurate predictions (covering all true positive annotations) while avoiding negative locations*.
> To further substantiate this point, we quantified the diversity of our proposed locations by counting the number of boxes that remain after applying non-maximum suppression (NMS) to an initial set of predictions for the same object in the same scene.
> Our results show that the predictions are diverse, with 82.5 boxes surviving the filtering stage out of the original 100 predictions.
> These results will be included in the revised version.
>
> [1] Wasserman, Navve, et al. "Paint by Inpaint: Learning to Add Image Objects by Removing Them First." (2024)
>
> [2] Zhao, Lirui, et al. "Diffree: Text-guided shape free object inpainting with diffusion model." (2024).
>
> [3] Haque, Ayaan, et al. "Instruct-nerf2nerf: Editing 3d scenes with instructions." ICCV (2023).
>
> [4] Shahbazi, Mohamad, et al. "InseRF: Text-Driven Generative Object Insertion in Neural 3D Scenes." (2024)

---

> ### Author Response · Authors · 2024-11-20
> **Initial response (2) 11/19**
>
> ### Case studies and analysis
>
> We plan to include out-of-domain results and experiments with multiple locations in the revised version.
> If there are additional aspects you would like us to address, please let us know.
>
> ### Decoupling instructions
>
> Could you please let us know what you mean exactly by "the authors should do some study on decoupling instructions".
> Is this about editing instructions?
> If so, in what sense should these be decoupled?

---

> > ### Author Response · Authors · 2024-11-22
> > **Response 11/22**
> >
> > ### Multiple placements in cluttered scenes
> >
> > We have uploaded a revision for our paper, showcasing some qualitative examples of the behavior of our location model when sampling multiple times for the same background image.
> > Our model is also capable of identifying multiple plausible locations in cluttered scenes, while avoiding arbitrary predictions.
> > These examples are shown in Figure 17 and 18 in the Appendix.
> > Additionally, we have added additional examples for out-of-domain results of our location model in Figure 19.
> > Lastly, we add failure cases in Figure 20, which show that if the location prediction is poor, the inpainting model can inadvertently occlude and remove existing objects.

---

### Official Review · Reviewer_MEvC · 2024-11-03

**Soundness:** 2
**Presentation:** 1
**Contribution:** 2
**Rating:** 3
**Confidence:** 4

**Summary:**

Generative models have become a powerful tool for image editing tasks, including object insertion. However, these methods often lack spatial awareness, generating objects with unrealistic locations and scales, or unintentionally altering the scene background. A key challenge lies in maintaining visual coherence, which requires both a geometrically suitable object location and a high-quality image edit. In this paper, They focus on the former, creating a location model dedicated to identifying realistic object locations. Specifically, They train an autoregressive model that generates bounding box coordinates, conditioned on the background image and the desired object class. This formulation allows to effectively handle sparse placement annotations and to incorporate implausible locations into a preference dataset by performing direct preference optimization.

**Strengths:**

They train an autoregressive model that generates bounding box coordinates, conditioned on the background image and the desired object class. This formulation allows to effectively handle sparse placement annotations and to incorporate implausible locations into a preference dataset by performing direct preference optimization.

**Weaknesses:**

- Novelty. The proposed method finetune the instruction-guided editing dataset and leverage the location model to generate the bbox, It has been appeared in previous mask-free inpainting work. What is the significance and value of using autoregressive models? What is the motivation of this work? I strongly think the motivation is weak.
- Experiment. the lack comparison with  brushnet.
- The ablation experiment using autoregressive and other models is missing.
- limiting the user's editing freedom.
- Generation with control signal such as depth and other sketch.

**Questions:**

See  weaknesses

---

> ### Author Response · Authors · 2024-11-20
> **Initial response 11/20**
>
> ### Clarification on the contributions
>
> We would like to provide major clarifications regarding the motivations and contributions of our work.
> This paper does not propose an image editing model nor to fine-tune a diffusion model to handle text instructions.
> In this paper, *we focus on developing a location model*, which can explicitly provide diverse candidates for plausible locations given an image.
> Whereas our paper showcases the use of our location models for image editing, these techniques are crucial for tasks requiring spatial awareness in general, as they enable more accurate and contextually appropriate object placement in various applications.
> While users can manually select locations in artistic editing, location models automate object placement in scenarios where manual intervention is impractical, such as in large scale data augmentation or object placement for augmented reality.
>
> Our location model itself is also fundamentally different from existing location models, as we take a generative approach, rather than classifying locations into plausible and implausible.
> We believe that this distinction allows our method to handle sparse annotations effectively and produce diverse location predictions, which are critical for improving object insertion tasks.
>
> ### On mask-free inpainting
>
> We are not entirely sure what you mean by mask-free inpainting models, could you clarify?
> We interpret it as "inpainting models that do not need a mask input", such as instruction-based ones.
>
> Among them, we believe it is worth to discuss Diffree [1], as this model explicitly predicts an object mask as part of the inpainting process.
> However, the mask is predicted for an object that is *already inserted in the scene* (precisely, in the latents) as a result of the diffusion process.
> The authors decode this mask only to enable seamless blending with the background.
> This strategy is fundamentally different than ours, as we predict the location of the edit *before the insertion takes place*.
> Crucially, we show in Figure 4 and Table 1 that our strategy is superior.
>
> ### Comparison against BrushNet
>
> BrushNet is an inpainting model that relies on user-provided locations.
> Rather than being a method for a fair comparison, it could represent a further choice model that can be paired with our location model (along with GLIGEN and PowerPaint in Section 4.4).
> We highlight that the PowerPaint model (v2 [2]) we use is already built on top of BrushNet.
> We will clarify this aspect and cite BrushNet in the revised version.
>
> ### Depth and Sketch Control Signals
>
> We are unsure what this comment refers to, and would appreciate if the reviewer can clarify their question.
> It seems to pertain to using ControlNets or T2I adapters to generate images conditioned on depth or sketch inputs.
> However, these conditions are not available for inserting new objects, as the object is not yet present in the scene, and inserting a depth map or sketch into a scene automatically is not a trivial task.
> Please let us know if this question has been addressed by our response or requires further elaboration.
>
> ### Ablations
>
> We perform ablations for DPO training and the choice of inpainting model in Section 4.4.
> If the reviewer points out which ablations they would have liked to see, we can comment.
>
>
> [1] Zhao, Lirui, et al. "Diffree: Text-guided shape free object inpainting with diffusion model." (2024).
>
> [2] https://github.com/open-mmlab/PowerPaint

---

### Official Review · Reviewer_b8Tg · 2024-11-03

**Soundness:** 3
**Presentation:** 3
**Contribution:** 3
**Rating:** 6
**Confidence:** 4

**Summary:**

This paper presents the novel task of generative location prediction for object insertion in images. Given an input backgorung image and an object categories, the proposed method predicts plausible locations for placing an object of the queried category in the backgorund image.

Specifically, authors propose training a conditional auto-regressive generative model, which predicts a spatial probablility distribution for the boundig boxes where the object can be placed. The proposed model is trained in two stages: first, the model is trained on two datasets using positive annotations of plausible locations for the objects. Then, a small dataset contatining both positive and negative annotations is used to further optimize the model using a preference optimizaiton technique.

The authors evaluate their proposed method in two setups:
1. locatoin modeling, where the model and its baselines are evaluated in their ability to predict plausible object locations,
2. Object insertion, where the model is compared to the baselines in their impact on image editing methods when used to create an object in an image.

**Strengths:**

- Location prediction for object insertion is a novel and challenging task, which has not been addressed in previous studies. The addressed task has multiple useful applications, and this work opens the door for furhter explorations of this problem.

 - The use of preference alignment in addition to the standard training of the autoregressive model is effective in improving the performance by exploiting available negative samples.

- Based on the provided evaluations, the proposed method is effective in providing reasonable locations for object inpainting methods compared to implicit location modeling done in currect editing models.

**Weaknesses:**

- The proposed approach heavily relies on annotated training datasets. Acquiring large dataset with rich annotations for such method is very difficult, considering many possibilities for annotations per image. My main concern is how well the proposed method generalizes beyond the current training data.

- I could not find any analysis on multiple predictions for the same object category and background image. It would be interesting to see how diverse and plausible are different predictions of the model for the same input.

**Questions:**

- The proposed method is first trained on PIPE dataset and then on OPA dataset (before preference alignment). Is there a reason the model is not trained jointly on the two datasets?

- As mentioned by the authors, the PIPE dataset could contain inpainting artifacts in many images. First of all, did the authors perform any filtering on such images? and secondly, did the authors analyze if the model tends to take a shortcut for its prediction by using the inpainting artifacts?

- Could the authors confirm if all the provided quantitave results for location modeling evaluation are performed on OPA dataset, where no inpainting is performed on the images?

---

> ### Author Response · Authors · 2024-11-19
> **Initial response 11/19**
>
> ### Inpainting artifacts in PIPE
>
> As mentioned, we found that some images in the PIPE dataset contain inpainting artifacts, resulting from the object removal procedure they went through.
> The cases for which such artifacts are visible are generally rare, but we believe some non-perceptible variations in pixel statistics might exist at a large scale.
> For instance, we empirically observed that Paint-by-Inpaint (trained on PIPE) consistently edits the same location in PIPE test images, even for scenes where multiple placements could be plausible.
> This behavior supports the existence of inpainting artifacts, despite the filtering stage performed by the authors of the dataset, and it is one of the reasons we evaluate on the OPA dataset as well.
>
> We believe that this issue could be problematic as it might trick the location model into learning meaningless shortcuts, simply identifying the artifact region as a plausible edit location without higher level or semantic scene parsing.
> Even worse, artifacts in the test set might over-reward this trivial behavior in the metrics, as ground-truth locations will coincide with them by construction.
>
> It is difficult to quantify this effect or filter PIPE images, as we do not know at which point the artifacts become harmful.
> In our experiments, we do not perform additional filtering of PIPE, to enable a fair comparison with the released Paint-by-Inpaint checkpoints.
> To ensure we work on images without inpainting artifacts, *we perform all quantitative experiments and analyses, with the exception of Table 1, on the OPA dataset*.
>
> ### Generalization beyond the training data
>
> It is true that our approach relies on richly annotated data, and this highlights a challenge shared by many object insertion methods.
> We agree that generalizing beyond our the training data (e.g., diverse object classes or scene images) is an important feature for location models.
> We will test and provide samples from our location model in the revised paper.
>
> ### Handling multiple targets
>
> We will add qualitative results that show diverse location predictions made by our location model.
> We have quantified diversity by counting the average number of bounding boxes that survive a phase of non-maximum suppression (NMS), with an IoU threshold of 0.7.
> NMS is a standard postprocessing in object detection, and it aims at filtering out predictions that overlap too much.
> Intuitively, a model predicting the same locations would result in a *lower* number of remaining boxes after NMS.
> On the OPA dataset, *we count on average 82.5 boxes after NMS, starting from a 100 predictions*, which showcases the diversity of our predictions.
>
> ### Training strategy
>
> The reason we relied on pretraining on PIPE and finetuning on OPA is merely because of their difference in scale (PIPE feature almost 900K training images, whereas OPA only around 1K).
> Whereas the scheme we employed seems more natural to us, joint training is also possible.
> We however believe that in that case extra care should be taken in rebalancing training examples from the two sources, to prevent PIPE from outweighting OPA.

---

> ### Author Response · Authors · 2024-11-22
> **Response 11/22**
>
> Thank you for the quick response.
>
> ### Multiple targets
>
> We have uploaded a revised paper that now includes qualitative samples showcasing our model’s ability to handle multiple targets as presented in Figure 17 and 18.
> Our model is capable of predicting diverse locations while ensuring they remain within plausible ranges.
>
> ### Generalization
>
> In Figure 19, we observe that *our model can generalize to placing certain object classes beyond the ones included in PIPE or OPA*, such as "pirate ship" or "flying car".
> We believe this zero-shot capability is inherited from the CLIP text encoder to producing representations of object classes, as it naturally assigns similar embeddings to known objects (e.g. ship, car).
> Nevertheless, the location model places the "flying car" on the ground, and unless trained on such classes, it is likely less suited for more artistic or complex out-of-domain classes.

---

> ### Comment · Reviewer_b8Tg · 2024-11-22
> **Feedback on the Rebuttals**
>
> I appreciate the authors’ responses to my questions. Some of my concerns, such as the diversity of multiple predictions and inpainting artifacts, have been resolved.
>
> Based on the additional results provided, I believe the method demonstrates promising outcomes but also exhibits several failure cases. For example, in Figure 19, the dog is located within the predicted regions for cars, and the new elephant is placed in place of the existing baby elephant. Similar behavior is observed in Figures 18 and 20. One weakness of the method, based on these results, appears to be its tendency to provide implausible predictions in cluttered scenes, as the authors have noted.
>
> There are also implausible predictions regarding the size of objects. For instance, in Figure 17, some of the bounding boxes predicted for the fire hydrant are disproportionately large given the actual expected size.
>
> These failure cases are indeed attributable to the challenging nature of the task. While the provided results show improvements over implicit spatial modeling in existing inpainting methods, they do not fully convince me of the effectiveness of the proposed method, particularly outside the domain of the limited training data. Therefore, I am currently inclined to maintain my score as it is.

---

> > ### Author Response · Authors · 2024-11-29
> > **Response 11/29**
> >
> > We are glad to hear that the additional clarifications, especially around multiple target predictions and inpainting artifacts, have helped address your concerns.
> >
> > We acknowledge that not all predictions are equally plausible.
> > However, this is a general problem for localization methods and instruction-finetuned models.
> > Determining where to place objects is not a solved problem, but our paper takes a step forward by focusing on improving the quality of locations, which has previously been overlooked.
> > As shown in Figure 6, we improve the true positive to false positive ratio (TPR:FPR) from 4.8:1 to 8.49:1, demonstrating a clear advantage in terms of reliability.
> > This suggests that *failures are far less frequent with our approach*, and we believe that this progress positions our paper as a meaningful contribution to the long-standing challenge of spatially aware object placement.

---

### Official Review · Reviewer_4jei · 2024-11-08

**Soundness:** 3
**Presentation:** 3
**Contribution:** 2
**Rating:** 6
**Confidence:** 4

**Summary:**

This paper presents a novel approach to object insertion in images, focusing on the generation of realistic object locations, which, when paired with an inpainting method, leads to visually coherent and accurate results.  This paper trains an autoregressive model that generates bounding box coordinates, conditioned on the background image and the desired object class, allowing to effectively handle sparse placement annotations and to incorporate implausible locations into a preference dataset by performing direct preference optimization.  Experiments show that this approach achieves better performance in location modeling and significantly outperforms instruction-tuned image editing models in object insertion tasks.

**Strengths:**

The paper presents a novel two-step process for object insertion, focusing on location modeling to improve the realism of object placement. Demonstrated superior performance in object insertion tasks, particularly in maintaining visual coherence.

**Weaknesses:**

Computational Cost: The addition of a location model might increase the overall computational burden of the system.
Data Dependency: While the model can work with sparse annotations, I think its performance remains heavily dependent on the quality of the dataset.

**Questions:**

Overall, I think the paper technically makes sense, but some questions need further discussion or resolution.

Q1: The paper pursues the accuracy of reasonable insertion target positioning, but for artistic creation, imagination needs to be infused. I am curious about the effect of inserting and editing imaginative objects. For example, what if the inserted object is a car with wings? Would the car be placed on the ground or in the air? Or could you provide other examples?

Q2: Can the method handle multiple targets? I am interested in whether the detection model has its own preferences. For instance, if the prompt is "add an apple to the top left and top right corners," how does the method in this paper deal with such a scenario?

Q3: I am not entirely sure about the motivation behind the application. Since the method in this paper first generates candidate positions and then performs inpainting to improve the reasonableness of the generated target, if the target position is so crucial, why not opt for manually drawing a candidate box on the background and then editing? This step is not complicated, and the position of the box should better align with the user's needs. My understanding is that this paper achieves automatic target box generation, but it adds a new model for generating boxes that comes with additional costs in terms of time, memory, and GPU resources. So, what extra advantages does this approach offer over manual box drawing?

---

> ### Author Response · Authors · 2024-11-19
> **Initial response 11/19**
>
> ### Motivation for location modeling
>
> In this work, we focus on predicting locations of new objects based on a class label alone.
> While manually specifying object locations remains feasible (in some cases, preferable) in artistic and creative applications, *a location model is particularly valuable for automated insertions*, where relying on interactions from the user becomes impractical due to scale requirements.
> An example is data augmentation applications: if we insert examples of specific object categories for training object detectors, it is likely that a large number of images need to be edited, likely in the order of the tens of thousands.
>
> We believe that location modeling is also a relevant problem for the broader research community, beyond image editing applications.
> Another example is augmented reality: placing virtual objects in a scene requires realistic location proposals, and cannot rely on humans providing the location in real time.
>
> In this paper, we focus on image editing as the main application, as the difference in location quality can clearly be observed in this context.
> We will clarify our contribution and emphasize the broader importance of location modeling in our revision.
>
> ### Computation overhead
>
> We have evaluated the overhead of our autoregressive location model in Section 5, lines 428-510.
> Given that current models for inpainting heavily rely on latent diffusion models, and require a decent number of forward passes for generating a single image, we observed that the overhead is minimal: predicting the location takes only *0.4 percent* of the time required to generate an image.
> We believe this to be a reasonable cost if the location model helps enhance the realism of the resulting image.
>
> ### Handling multiple targets
>
> Our method can predict multiple target locations, but in independent sampling steps.
> We made this choice (independence) as it significantly simplifies the training of the autoregressive model: if we would need to autoregressively decode more than one bounding box, we should during training establish an arbitrary order among them, that we thought might hinder the optimization.
> As a result, the model cannot currently handle prompts such as "add an apple to the top left and top right corners".
> We would handle this scenario in two separate edits: first editing the image to add an apple to the top left corner, and then to add another to the top right.
>
> In general, it is difficult to determine whether the location model has specific preferences or if the chosen locations are simply the most plausible based on the scene's structure and content.
> We do provide example outputs in Figure 8, and Figure 15 in the Appendix, which show that our location model predicts boxes of different sizes and different locations, whereas other methods often tend to add objects in similar locations.
> Nevertheless, we agree that investigating potential biases in location models could offer valuable insights and serve as a promising direction for improving their performance.
>
> ### Requirement for high-quality data
>
> We acknowledge that our approach relies on a publicly available high-quality dataset for modeling locations, which can be difficult to collect.
> We believe that this currently represents the main challenge inherent to *all existing location modeling studies*, as precise annotations are critical for meaningful results.
> Unsurprisingly, some of our baselines [1,2] focus on curation of a synthetic paired dataset.
> We hope to explore ways to leverage larger or web-scale datasets in the future.
>
> [1] Wasserman, Navve, et al. "Paint by Inpaint: Learning to Add Image Objects by Removing Them First." (2024)
>
> [2] Zhao, Lirui, et al. "Diffree: Text-guided shape free object inpainting with diffusion model." (2024).

---

> > ### Comment · Reviewer_4jei · 2024-11-20
> > **Comments to the Initial response 11/19**
> >
> > Thanks for the author's responses.
> >
> > 1. No response to my first question about the artistic creation.
> > 2. "Computation overhead":  what are the memory storage (model size) and GPU resource requirements?

---

> > > ### Author Response · Authors · 2024-11-22
> > > **Response 11/22**
> > >
> > > Thank you for the quick response.
> > >
> > > ### Artistic creation
> > >
> > > We have now uploaded the revised paper, which includes cases where object classes unseen during training are inserted (Figure 19).
> > > However, we see that the location model does not place "flying cars" in the sky, but predicts this class will be on the ground.
> > > We ascribe this behavior to the nature of our training data, where the closest class according to the CLIP text encoder is likely to be "car'", and this class will typically be on the ground as well.
> > > To enable more artistic or out-of-domain classes, training on similarly diverse classes is likely required.
> > >
> > > ### Computational overhead
> > >
> > > Our location model consists of 411 million parameters which only requires 2.05 GB.
> > > Inference takes 3.98 GB of VRAM, which means that the model can easily run on consumer-grade hardware.
> > > We have added a more detailed discussion of computational cost in Section D.1.
> > > We hope this addresses your concerns about memory storage and GPU resource requirements.

---

> > > > ### Comment · Reviewer_4jei · 2024-11-24
> > > > **Comments to the  response 11/22**
> > > >
> > > > Thanks for the author's response. This solves most of my concerns. Although there may be trade-offs in terms of computing resources and imagination, they are tolerable, so I will maintain my rating of 6 (marginally above acceptance).

---

### Official Review · Reviewer_U56Q · 2024-11-09

**Soundness:** 2
**Presentation:** 2
**Contribution:** 3
**Rating:** 5
**Confidence:** 2

**Summary:**

The paper focuses on proposing a location model to identify realistic object location in an image. This model is then further combined with the inpainting module is used to outperform SOTA instruction tuned models and location modeling baselines.

**Strengths:**

1. The paper is decently written paper, but it can still improve it's clarity.
2. The proposed experiments and the use of potive and negative locations in conditioning the model seems interesting.

**Weaknesses:**

1. First and foremost the paper specifies a model for localisation using an autoregressive model. One simple question, can the model use a mask using a model like segment anything model (SAM) to localise the object and simply use the inpainting model.
2. Second make the equations and it's more clear, such as equation 3, the reviewer is assuming it to be the "order".
3. The alternative inpainting models such as SmartBrush, LeftRefill etc, are not well explored.

**Questions:**

Please refer to the questions on the weakness and try to address those.

---

> ### Author Response · Authors · 2024-11-19
> **Initial response 11/19**
>
> ### Can't you use SAM?
>
> While object detectors or instance segmentation models such as SAM are well-suited for locating *existing objects*, they are not suitable for identifying where *new objects* could naturally fit within the scene, which is the focus of our work.
> A potential use of recognition models for image editing would be to detect and then replace existing objects, which does not align with our goal.
> Our goal is to automatically insert objects into scenes in new locations, without human feedback or interaction, which has use cases in XR and data augmentation.
> A dedicated location model determines unoccupied areas, and we are able to place new objects without altering existing ones.
>
> ### Including alternative inpainting approaches
>
> Thanks for pointing out additional inpainting models such as SmartBrush and LeftRefill.
> We used GLIGEN and PowerPaint as inpainting methods in our work, as GLIGEN is well-established work, and PowerPaint is very recent.
> We agree that including even more models would strengthen the conclusions of the paper.
> However, implementation and trained models for SmartBrush are not open-sourced, which makes it hard to use.
>
> LeftRefill is a reference-based inpainting method, which inpaints a (right) target image based on a (left) reference image.
> In this case, the user provides the mask, and the two images should match closely.
> In our case, there is no reference image, and we do not have a user-specified mask.
>
> Since inpainting models typically rely on users to provide the location, they could potentially benefit from the accurate and plausible locations generated by our location model.
> For these reasons, we chose GLIGEN and PowerPaint as main inpainting methods.
> We will cite other inpainting models in the revised paper to clarify our choice.
>
> ### On Equation 3 clarification
>
> Although the equation reports a standardized DPO formulation, we agree that its interpretation could be clarified within the text.
> We shall improve its readability in the final revision.
> Could you please clarify what you mean with the comment about the "order"?

---

> > ### Author Response · Authors · 2024-11-22
> > **Response 11/22**
> >
> > We have now uploaded the revised version of our paper, we have cited the suggested inpainting methods and provided additional context for Equation 3.
> > We hope these updates address the concerns and provide further clarity.

---

> > > ### Comment · Reviewer_U56Q · 2024-11-25
> > >
> > > Thanks to the authors for the response and adding more clarity. The authors did answer the points but the first point can be clarified better. For equation 3, by order I meant that it is the preference order, that is clarified. Also looking into the reviews by MEvC and QUkD I would be maintaining my score.

---

> > > > ### Author Response · Authors · 2024-11-29
> > > > **Response 11/29**
> > > >
> > > > Thank you for the thoughtful feedback and for taking the time to review our revisions.
> > > >
> > > > To address the feedback more thoroughly, we will add further clarifications distinguishing between models that focus on recognizing existing elements (e.g., SAM) and those that identify locations for new elements to be inserted (i.e., location models).
> > > > We addressed various points from reviewer QUkD as well, and clarified the limits and merits of the method.
> > > > *We believe that this is what led to a positive shift to their scores*, and hope that our response to their points also addresses some of your remaining concerns.
> > > >
> > > > ---
> > > >
> > > > If any further questions arise or clarifications are needed, we'd be happy to address them.
> > > > Otherwise, we hope the revisions meet your expectations and look forward to hearing your thoughts.

---

### Author Response · Authors · 2024-11-19

We would like to sincerely thank the reviewers for their valuable feedback and thoughtful questions.
The answers we provide today are intended to kickstart the discussion, and we warmly encourage further engagement and dialogue.
We will update the paper at a later stage to make sure all updates are included in the same revision and avoid confusion.

---

> ### Author Response · Authors · 2024-11-22
> **General Comment 11/22**
>
> Dear reviewers, **we have uploaded a revised version of our paper** to further address your comments.
> The new version comprises with the following revisions (marked with red colored text):
> * We added clarifications to Eq.3 [U56Q].
> * We added additional measurements and considerations about the computational cost in Section D.1 [U56Q].
> * We highlight the diversity of our predicted locations in Figure 17 and Figure 18, where our location model is generates multiple plausible locations for a single sample.
> These results align with our quantitative analysis in Section E, where we run a standard non-maximum suppression (NMS) filtering to 100 predictions with a threshold of 0.7, resulting in 82.5 unique boxes [b8Tg, QUkD].
> * We showcase the model’s generalization to unseen object classes beyond those in the dataset in Figure 19.
> We additionally show in the same figure some artistic insertions (rightmost column), which were also not part of the training dataset [4jei].
> * We show failure cases in Figure 20, such as when objects are difficult to integrate into the scene (e.g., adding an airplane to a scene with no visible sky) [QUkD].
>
> We hope these additions to the paper help reviewers better understand our proposed model’s capabilities and limitations.
> We have also addressed specific details in the individual comments for further clarity.
> We welcome any further comments and feedback that reviewers might have upon assessment of the new revision.

---

### Meta-Review · Area_Chair_VbWb · 2024-12-15

**Metareview:**

This paper receives 3 negative ratings and 2 positive ratings. Although the paper has some merits, e.g., technical motivation and soundness, the reviewers pointed out a few critical concerns about 1) dependency on the training dataset, 2) technical novelty and capability like handling multiple objects, 3) generated artifacts and failure cases. After taking a close look at the paper, rebuttal, and discussions, the AC agrees with reviewers' feedback and hence suggests the rejection decision. The authors are encouraged to improve the paper based on the feedback for the next venue.

**Additional Comments On Reviewer Discussion:**

In the rebuttal, some of the concerns like computing resource are addressed by the authors. However, during the post-rebuttal discussion period, the reviewer QUkD is not convinced about a few limitations of the work, e.g., generated image quality, dependency on a large training dataset, multi-object handling. The AC took a close look and agrees that the results indeed contain several failure cases that are not geometrically reasonable (also pointed out by the reviewer b8Tg who gave the positive rating). Along with the consideration of limited technical novelty of using the location model (but still limited in geometry), the AC agrees with the reviewers that these issues should be significantly improved in the manuscript, which still requires a good amount of effort to make the paper ready for publication.

---

### Decision · Program_Chairs · 2025-01-22

Reject